# *Bifidobacterium* response to lactulose ingestion in the gut relies on a solute-binding protein-dependent ABC transporter

Keisuke Yoshida [1]✉, Rika Hirano[2,3], Yohei Sakai[4], Moonhak Choi[5], Mikiyasu Sakanaka [5,6], Shin Kurihara[3,6], Hisakazu Iino[7], Jin-zhong Xiao [1], Takane Katayama[5,6] & Toshitaka Odamaki[1]

This study aims to understand the mechanistic basis underlying the response of *Bifidobacterium* to lactulose ingestion in guts of healthy Japanese subjects, with specific focus on a lactulose transporter. An in vitro assay using mutant strains of *Bifidobacterium longum* subsp. *longum* 105-A shows that a solute-binding protein with locus tag number *BL105A_0502* (termed LT-SBP) is primarily involved in lactulose uptake. By quantifying faecal abundance of LT-SBP orthologues, which is defined by phylogenetic analysis, we find that subjects with $10^7$ to $10^9$ copies of the genes per gram of faeces before lactulose ingestion show a marked increase in *Bifidobacterium* after ingestion, suggesting the presence of thresholds between responders and non-responders to lactulose. These results help predict the prebiotics-responder and non-responder status and provide an insight into clinical interventions that test the efficacy of prebiotics.

[1] Next Generation Science Institute, RD Division, Morinaga Milk Industry Co., Ltd., Zama, Japan. [2] Research Institute for Bioresources and Biotechnology, Ishikawa Prefectural University, Nonoichi, Japan. [3] Biology-Oriented Science and Technology, Kindai University, Kinokawa, Japan. [4] Food Ingredients and Technology Institute, RD Division, Morinaga Milk Industry Co., Ltd, Zama, Japan. [5] Graduate School of Biostudies, Kyoto University, Kyoto, Japan. [6] Faculty of Bioresources and Environmental Sciences, Ishikawa Prefectural University, Nonoichi, Japan. [7] Life Science for Living System, Graduate School, Showa Women's University, Tokyo, Japan. ✉email: keisuke-yoshida826@morinagamilk.co.jp

The human gut microbiota is one of the most densely populated ecosystems in the body. Recently, the gut microbiota has been revealed to have a strong association with the development of human diseases, such as colon cancer[1], obesity[2] and diseases related to infection by pathogenic bacteria[3,4]. This indicates that the gut microbiota is a potential target for the treatment and prevention of certain diseases. *Bifidobacterium* is considered to have a positive effect on human health and is effective in the prevention and/or treatment of conditions such as obesity[5], allergic symptoms[6], and Alzheimer's disease[7]. In addition to oral treatment with probiotic bifidobacteria, some oligosaccharides, such as lactulose, galactooligosaccharides (GOS) and fructooligosaccharides (FOS), are also expected to act as prebiotics that increases the bifidobacterial population in the gut[8,9]. Lactulose is a simple disaccharide composed of fructose and galactose that is widely used as a prebiotic, as well as a laxative[10]. Many studies have revealed the capability of lactulose to promote the increase in abundance of *Bifidobacterium* in the human gut[11–14]; therefore, this compound is widely applied in the food and pharmaceutical industries[11,13,14].

Recently, the concept of precision medicine and nutrition has received much attention in the field of life sciences, due to significant inter-individual variation in response to treatment. In clinical settings, studies have indicated that the commensal microbiota is associated with the efficacy of immune checkpoint blockade therapy in melanoma patients[15–17]; some gut microbes, such as *Bifidobacterium*, *Faecalibacterium*, and *Akkermansia*, were found to be enriched in the patients clinically responding to therapy. Previous studies show that response to diet is also dependent on the gut microbiota. Bäckhed's group reported that subjects with improved glucose metabolism in response to barley kernel supplementation exhibited increased *Prevotella* abundance in their gut microbiota, supporting the importance of personalised approaches to improve metabolism[18]. It is well known that the effect of prebiotics in promoting the growth of *Bifidobacterium* in the gut differs among subjects[8,19]. In a previous clinical study, which was a randomised, double-blind, placebo-controlled crossover trial, we observed that 2 g of lactulose ingestion per day for two weeks increased the number of *Bifidobacterium* cells in the faeces of 49 healthy Japanese females with statistical significance (2.2 (2.9) fold increase, geometric mean (standard deviations))[9]. However, the response of *Bifidobacterium* varied among individuals. The fold change in cell number of *Bifidobacterium* was between 0.38 to 48.11, suggesting the existence of responders and non-responders to lactulose ingestion. We speculate that this variation may be caused by the difference in lactulose assimilation ability of the bifidobacteria residing in individual intestines. Therefore, if the transporter to incorporate

prebiotics can be clearly determined, responders for each prebiotic ingredient could be identified by analysing targeted genes of the gut microbiota. Solute-binding proteins (SBPs) of ATP-binding cassette (ABC) transporters have been identified to be involved in the uptake of oligosaccharides by *Bifidobacterium*[20–23]; however, the transporter for lactulose incorporation remain largely elusive[24,25].

In this study, we have identified the SBP possibly responsible for lactulose uptake by *Bifidobacterium longum* subsp. *longum* 105-A, as a representative of lactulose-utilising strains. We also found that high inter-individual variability in response to lactulose ingestion can be explained by the abundance of the transporter gene. The SBP we have identified could be a potential indicator to differentiate between lactulose responders from non-responders in clinical settings.

## Results

**ABC transporters involved in lactulose incorporation.** To identify a key gene responsible for the utilisation of lactulose, we first performed a BLASTP search against the genome of *Bifidobacterium longum* subspecies *longum* (*B. longum* subsp. *longum*) 105-A, a lactulose-utilising strain, using *Bal*6GBP (*BALAC_RS02405*) as a query sequence. *Bal*6GBP has been shown to bind to several β-galactosides, including lactulose[25]. Consequently, three adjacent genes (*BL105A_0500*, *BL105A_0501*, *BL105A_0502*), all of which encode the SBP of an ABC transporter previously assumed to be involved in the uptake of GOSs[20,22,24], were identified as a *Bal*6GBP homologue (Fig. 1). The amino acid identities of the three homologues to *Bal*6GBP are 59.1, 73.7, and 51.2%, respectively. (Fig. 1 and Supplementary Table 1). To determine whether these genes were involved in lactulose import in *B. longum* subsp. *longum* 105-A, we inactivated the respective genes by a single cross-over insertion and evaluated the growth of the three mutants on lactulose. The difference in growth between the wild-type strain and three mutants was indistinguishable in a medium supplemented with lactose as the sole carbon source (Fig. 2a). In contrast, in a medium with lactulose as the sole carbon source (mMRS-L), the *BL105A_0502* mutant showed impaired growth and lowered acetic acid production compared to the other strains (Fig. 2b, Supplementary Fig. 1). We, therefore, complemented the *BL105A_0502* gene by plasmid and evaluated the growth of the strain on lactulose. As expected, the complemented strain showed restored growth comparable to the wild-type strain (Fig. 2c, d). These data indicated that the SBP encoded by the *BL105A_0502* gene was responsible for the uptake of lactulose.

**Subtype classification of *BL105A_0502* homologues.** In a previous clinical study, lactulose ingestion was shown to increase the

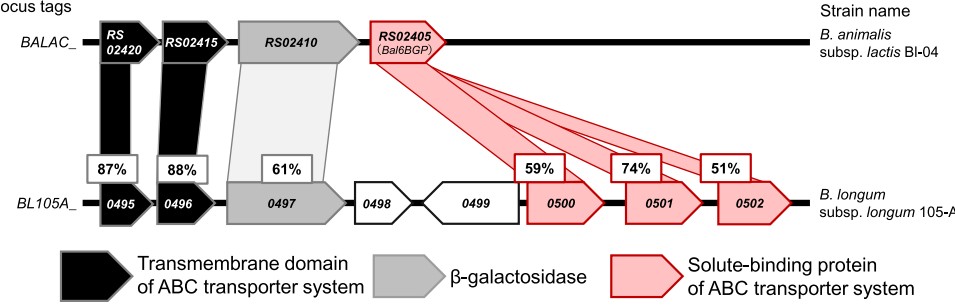

**Fig. 1 Comparison of the genomic structure of *B. animalis* subsp. *lactis* Bl-04 at the *Bal*6BGP locus with the corresponding region of *B. longum* subsp. *longum* 105-A.** Solid arrows indicate open reading frames (ORFs) with their lengths proportional to the polypeptide chain lengths. The locus tag numbers are indicated inside the arrows. The amino acid identity (%) between the homologues is shown. The genes coding for transmembrane component, β-galactosidase, and SBP are coloured in black, grey, and light red, respectively.

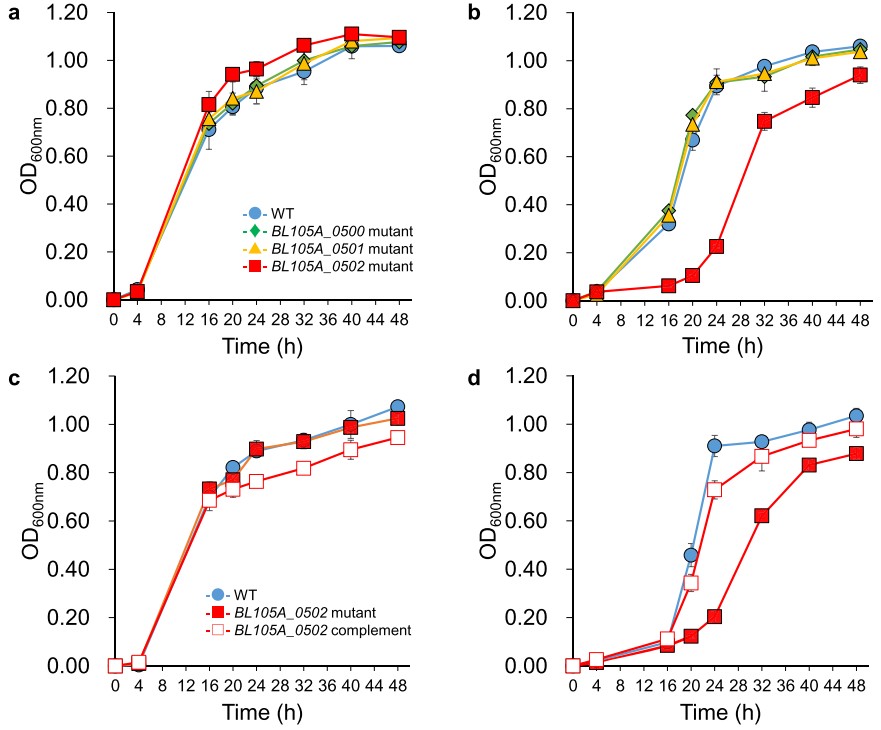

**Fig. 2 Growth profiles of *B. longum* subsp. *longum* 105-A wild-type, insertional mutants and a complemented strain.** Each strain was cultured in modified MRS medium supplemented with lactose (**a**)(**c**) or lactulose (**b**)(**d**) as the sole carbon source. OD$_{600nm}$ values were determined at the indicated time points. The presented data are the mean ± SD of at least three independent assays.

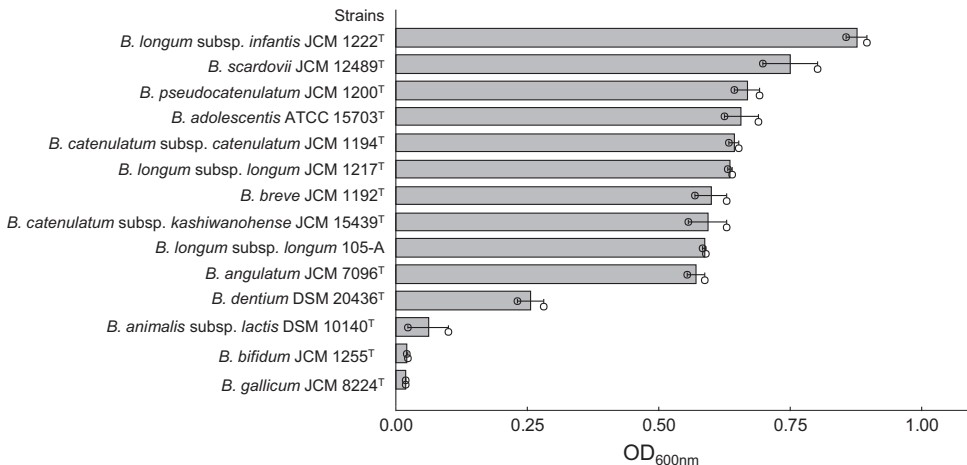

**Fig. 3 Growth capability of 14 strains belonging to 13 different *Bifidobacterium* species/subspecies in modified MRS medium supplemented with lactulose as the sole carbon source.** Strains with an OD$_{600nm}$ value greater than 0.3 after 24 h incubation was classified as a lactulose-assimilating phenotype (see Fig. 2b). The presented data are the mean ± SD of at least two independent assays.

abundance of four *Bifidobacterium* taxa in the faeces of forty-nine females: *B. catenulatum* group, *B. longum* group, *B. adolescentis* and *B. breve*[26]. We, therefore, examined the growth of 14 strains belonging to 13 different *Bifidobacterium* species/subspecies on mMRS-L. Among the tested strains, ten belonging to human residential *Bifidobacterium* (HRB) species/subspecies, which include the four strains mentioned above[26], showed marked growth (Fig. 3). Interestingly, *B. animalis* subsp. *lactis* DSM 10140, which harbours Bal6GBP, an SBP with low affinity to lactulose ($K_d = 4.9 \times 10^2$ μM)[25], did not grow on lactulose. We then performed a BLASTP search against the genomes of the lactulose-assimilating strains to investigate whether the

BL105A_0502 homologue (hereafter referred to as LT-SBP) was conserved. As expected, LT-SBP homologues were distributed in all of the lactulose-assimilating strains (Supplementary Fig. 2). A phylogenetic tree constructed based on the amino acid sequences revealed that the LT-SBPs were divided into four subtypes (Fig. 4): *B. adolescentis* subtype (BBKW_RS02570, BBCT_RS02630, BBPC_RS02630, BAD_RS02525), *B. longum* subtype (BBBR_RS02325, BLLJ_RS02295, BLIJ_RS10460), *B. angulatum* subtype (BBAG_RS07445) and *B. scardovii* subtype (BBSC_RS03380).

An additional BLASTP search against all publicly available genomes of *Bifidobacterium*, except for unidentified species,

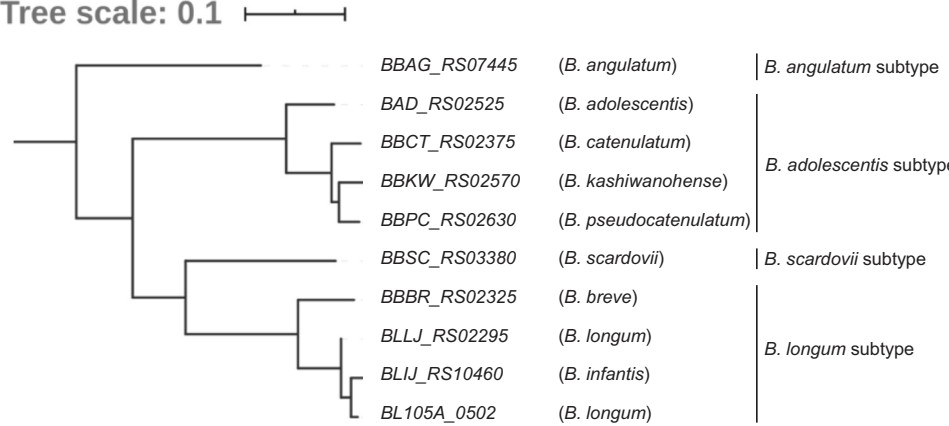

**Fig. 4 Phylogenetic tree of *BL105A_0502* homologues present in lactulose-assimilating *Bifidobacterium* genomes.** The tree was constructed using the amino acid sequences of *BL105A_0502* homologues (e-value threshold <$10^{-100}$) from 10 *Bifidobacterium* genomes with the lactulose-assimilation phenotype (see Fig. 3). The locus tag number of each *Bifidobacterium* genomes (parenthesised) and the phylogeny-based SBP subtype classification are shown.

(1033 genomes in total) showed that almost all strains belong to 57 of 77 bifidobacterium species possessed the LT-SBP homologue (Supplementary Table 2). Another BLASTP search against the refseq protein database showed that a few species of other genera were also predicted to possess the LT-SBP homologous gene (Supplementary Table 3).

**Effect of lactulose ingestion is associated with faecal LT-SBP abundance.** Based on the above results, we hypothesised that LT-SBPs might serve as an indicator that explains the observed inter-individual variance in response to lactulose ingestion in the above-mentioned clinical study. Real-time PCR using four primer pairs designed for each LT-SBP subtype (Supplementary Table 4) revealed that LT-SBP was detected in 46 of 49 subjects (94%) before ingestion, and the total copy number of the four SBP gene subtypes upon lactulose ingestion with statistical significance after ingestion (2.2 (3.0)-fold, $p = 5.7 \times 10^{-6}$, tow-tailed paired student's *t*-test, Fig. 5a). Overall, with a significant positive correlation in fold changes between *Bifidobacterium* and LT-SBPs (Pearson correlation coefficient, $r = 0.93$, $p = 7.1 \times 10^{-22}$, Fig. 5b), almost all LT-SBP genes were presumed to be attributable to *Bifidobacterium*.

The fold increase in *Bifidobacterium* abundance was significantly higher in subjects with a total SBP gene abundance within the range of $10^7$–$10^9$ copies per gram of faeces before ingestion (moderate LT-SBP group, $n = 15$) than in the subjects outside of that range (less than $10^7$ or more than $10^9$ copies per gram, $n = 4$ or $n = 30$, respectively) (Fig. 5c). It is interesting to note that all of the subjects with more than a fivefold increase were in the moderate LT-SBP group (indicated in red in Fig. 5d). These results imply that there are both upper and lower thresholds between lactulose responders and non-responders, and we can, therefore, predict subjects who are potential "high responders" based on the copy number of LT-SBPs in faeces before lactulose ingestion.

**LT-SBP distribution in the guts of healthy subjects.** To predict the rate of potential "high responders" in the Japanese cohort, we assessed the prevalence of LT-SBPs by real-time PCR using four primer pairs. Among the 394 faecal samples examined, the *B. longum* subtype LT-SBP showed the highest prevalence (304 tested positive, 77%), followed by the *B. adolescentis* subtype LT-SBP (65%). The other two subtypes were either scarce or not detected (*B. angulatum* subtype: 0.76%; *B. scardovii* subtype: 0%).

In total, the LT-SBP gene was detected in 89% of Japanese subjects. The prevalence of subjects within the moderate LT-SBP group ($10^7$–$10^9$ copies per gram of faeces), who were assumed to be high responders to lactulose, was 49.2% (Supplementary Table 5).

Finally, we assessed the distribution of LT-SBP in gut microbiota using metagenomic datasets analysed previously in two different cohorts[27,28]. The LT-SBP homologue was detected in 89.7% of individuals in one cohort (Japanese) and 29.0% in another cohort (Danish) (Supplementary Table 6).

## Discussion
Prebiotics are believed to have a major impact on the gut microbiota composition, but the efficacy is considered to be individual-dependent and unpredictable. In this study, we found that the increase in the abundance of bifidobacteria upon lactulose ingestion was dependent on the number of LT-SBPs, which was identified to be a primary lactulose transporter by gene disruption and complementation study.

We identified three SBP genes, namely, *BL105A_0500*, *BL105A_0501* and *BL105A_0502*, as homologues of *Bal*6GBP (originating from *B. animalis* subsp. *lactis* Bl-04), which has been reported to have a slight binding affinity for lactulose ($K_d$ of $4.9 \times 10^2 \, \mu$M)[25]. Our efforts to establish the three single SBP mutants of *B. longum* subsp. *longum* 105-A indicated that *BL105A_0502* plays an important role in lactulose utilisation, although *BL105A_0501* was the top hit against *Bal*6GBP. Considering that *B. animalis* subsp. *lactis* DSM 10140 had the same SBP (which was the *Bal*6GBP homologue, id = 100% by BLASTP search) but failed to grow in mMRS-L, the *BL105A_0501* homologue may not be as actively involved in lactulose incorporation as expected.

Another report also indicated that the expression of the *Bbr_0530* gene, which is a *BL105A_0501* homologue in *B. breve* UCC2003, was upregulated in a medium containing lactulose as the sole carbon source[29]. However, not all cases of this gene expression reflect this function. Sotoya et al.[20] reported that an SBP protein (*BBBR_RS08090*) in *B. breve* YIT 4014 was involved in 3′-galactosyllactose (GL) utilisation, but SBP gene expression was not upregulated when cells were cultured in a medium containing 3′-GL as the sole carbon source. Considering the delayed, but not completely inhibited growth of the *BL105A_0502* mutant in mMRS-L (Fig. 2b), other mechanisms to incorporate lactulose must exist, such as the *lacS* permease, which was previously reported as a symporter for lactose and lactulose in *B.*

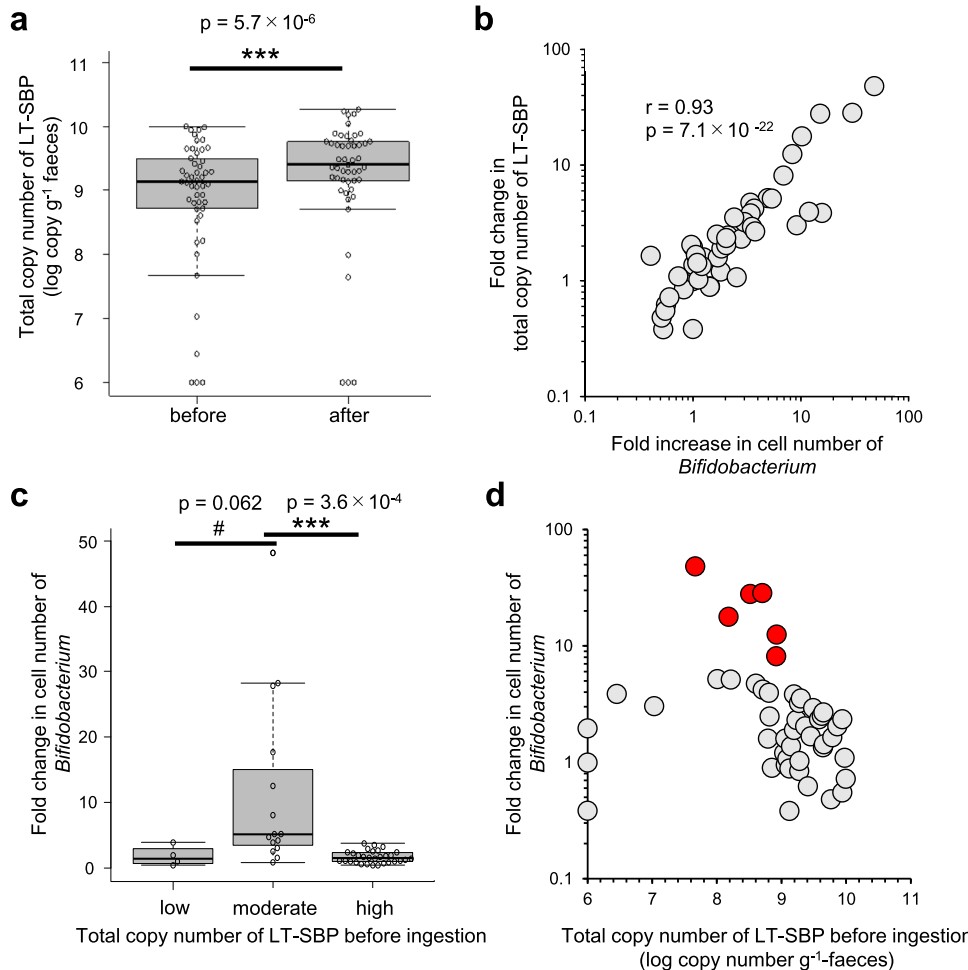

**Fig. 5 Association between the fold change in *Bifidobacterium* abundance and the total copy number of the LT-SBPs in faecal samples after ingestion** (*n* = 49). **a** Total copy numbers of the LT-SBP genes in faecal samples were determined by real-time PCR and compared before and after lactulose ingestion (two-tailed paired *t*-test). **b** Scatter plot of the individual samples based on the fold change in total copy number of the LT-SBPs and faecal *Bifidobacterium* cell number after lactulose ingestion (Pearson correlation coefficient). **c** Varied response to lactulose ingestion among the subjects is associated with the faecal copy number of LT-SBP genes before the trial. #p refers to <0.1, ***p < 0.001, Tukey's test. The box plots feature the median (centre line), upper and lower quartiles (box limits), 1.5× the interquartile range (whiskers), and outliers (points). **d** Scatter plot of individual samples based on the faecal copy number of LT-SBP genes before lactulose ingestion and the fold change in faecal *Bifidobacterium* cell number after lactulose ingestion. Red dots indicate subjects with a more than fivefold increase of *Bifidobacterium* after lactulose ingestion.

*breve* UCC2003[24]. Even so, only the *BL105A_0502* mutant did not grow well in mMRS-L in our study (Fig. 2a, b), and the BL105A_0502-complemented strain displayed recovered growth in the same medium (Fig. 2c, d). Therefore, we infer that *BL105A_0502* plays a primary role in lactulose utilisation.

The *BL105A_0502* (but neither *BL105A_0500* nor *BL105A_0501*) homologue was located in the genomes of all the lactulose utilisers (Supplementary Fig. 2), but not all strains possessed the full gene set of the ABC transporter (e.g., *B. longum* subsp. *infantis* JCM 1222, *B. adolescentis* ATCC 15703, see Supplementary Fig. 2). These orphan SBPs might be able to use membrane components encoded at different loci in the genomes, as suggested by Chen et al.[30]

Interestingly, several *Bifidobacterium* species/subspecies encode more than two different LT-SBP paralogues at the locus (Supplementary Fig. 2). This could allow these species/subspecies to utilise a wide range of galacto-series prebiotic sugars, which such as GOSs, lactulose, and plant-derived galactan degradants. The presence of three paralogues in the genome of *B. longum* subsp. *longum* may explain the distribution and persistence of this subspecies in subjects of different ages, ranging from infants to

centenarians[31–33]. The recent reports by Matsuki et al also revealed that *B. breve* utilises 3′-GL, 4′-GL, and 6′-GL via three distinct but function-overlapping ABC transporters, of which one is *BL105A_0501* homologue[20,21]. The findings also suggest the adaptation of this species to different galacto-series prebiotics.

We found that the fold increase in *Bifidobacterium* was prominent in the subjects with $10^7$–$10^9$ copies of the total SBP genes per gram of faeces before ingestion (Fig. 5c). However, the change after lactulose ingestion was inconspicuous in the subjects who had more than $10^9$ copies of LT-SBPs (per gram of faeces). This was possible because of the dilution effect, in which limited lactulose availability could mask the response of bifidobacteria to the prebiotic. Nevertheless, lactulose may be consumed by these bacteria and metabolised to produce short-chain fatty acids, such as lactic acid and acetic acid, which have been reported to be beneficial metabolites that regulate the colonic environment improve immune function, and prevent obesity[34,35]. In contrast, the abundance of *Bifidobacterium* was not significantly increased in subjects harbouring fewer than $10^7$ copies of the LT-SBP genes. Assumed that the LT-SBP gene is present at one copy in the genome and that the total bacterial cell number is $10^{12}$ in a gram

of faeces, the abundance of lactulose-assimilating *Bifidobacterium* in the guts of these individuals is estimated to be 0.001%. Therefore, it is not likely that the residing *Bifidobacterium* can gain access to lactulose before being foraged by other gut microbes which possess the LT-SBP homologue (shown in Supplementary Table. 3) or with moderate preference to lactulose. However, further investigation is needed to reveal the difference in molecular mechanisms among the three subject groups (harbouring low, moderate or high numbers of LT-SBPs).

Based on an observational study, real-time PCR analysis showed that bifidobacteria with LT-SBPs were widely distributed in Japanese subjects. Among these subjects, the prevalence of moderate-LT-SBP level subjects, who were predicted to be high responders to lactulose ingestion, was 49.2% (Supplementary Table 5). These results imply that approximately half of the Japanese individuals are potential responders to lactulose ingestion. Further data mining in a separate Japanese cohort showed that the LT-SBP prevalence was 89.7%, which almost the same as that in our observational study (89%). On the other hand, the metagenome data in the Danish cohort indicated a lower prevalence of the LT-SBP (29%). It has been reported that the relative abundance of *Bifidobacterium* is significantly higher in the Japanese than in the Danish[27,28]. The LT-SBP distribution, therefore, seems to be related to the *Bifidobacterium* abundance in each nation. These results imply that this prebiotic might be especially effective for the Japanese though further investigation with the appropriate number of subjects is needed.

In this study, we identified one of the transport systems involved in lactulose uptake in bifidobacteria. Furthermore, we found that the number of LT-SBPs could be a marker for predicting subjects with the potential for a marked increase in bifidobacterial abundance upon lactulose intake. We believe that the knowledge acquired from such findings will lead to precision nutrition and evidence-based intervention in the near future.

## Methods

**Faecal DNA collection**. The faecal DNA samples analysed in this study were obtained in our previous two studies that include healthy 49 females (18–31 years old) recruited in lactulose ingestion study[9] and 394 Japanese (20–104 years old) in a cross-sectional study[31,36]. The former clinical study was conducted to investigate the effect of lactulose ingestion on the defecation frequency of females at Showa Women's University (Tokyo, Japan) between May and December 2017. The study protocol was approved by the Institutional Review Board of Showa Women's University and is registered with the University Hospital Medical Information Network Clinical Trials Registry (No. UMIN000027305). The latter observational study was approved by the ethics committee of Kensyou-kai Incorporated Medical Institution (Osaka, Japan). All participants provided their written informed consent and were conducted in accordance with the principles of the Declaration of Helsinki.

**Bacteria and culture conditions**. The *Bifidobacterium* strains shown in Fig. 3 were grown anaerobically in 200 µL of modified MRS medium (Supplementary Table 7) with 1% (w v⁻¹) lactose (95% purity, Nacalai Tesque, Inc., Kyoto, Japan) or lactulose (MLC-97, a commercially available product of Morinaga Milk Industry Co., Ltd (Tokyo, Japan) with 97% purity, as determined by HPLC using a refractive index detector) as the sole carbon source at 37 °C for up to 24 h in 96-well tissue culture plates (Corning Incorporated, NY, USA). The growth curve was monitored by measuring the optical density at 600 nm ($OD_{600\ nm}$) at 0, 4, 16, 20, 24, 32, 40, and 48 h using a plate reader (Tecan Group Ltd., Zürich, Switzerland) in an anaerobic chamber (Ruskinn Technology, Ltd., Bridgend, UK). An $OD_{600nm}$ greater than 0.3 was judged as positive growth in the medium. An acetate assay kit (Bioassay Systems, Hayward, CA, USA) was used for measuring the concentration of acetic acid in the culture.

**Mutant construction**. The insertional mutation of SBP genes was carried out by single crossover recombination using a suicide plasmid as described previously[37]. *Escherichia coli* DH5α and an In-Fusion cloning HD kit (Clontech Laboratories, Inc., Mountain View, CA, USA) were used for DNA manipulation. Briefly, the internal regions of *BL105A_0500*, *BL105A_0501*, and *BL105A_0502* genes were amplified by PCR from the genomic DNA of *B. longum* subsp. *longum* 105-A[38] and ligated with BamHI-digested pBS423 fragment containing pUC *ori* and the

spectinomycin resistance gene[39]. The resulting suicide plasmids were independently introduced into *B. longum* subsp. *longum* 105-A by electroporation and respective insertional gene mutants were subsequently selected on Gifu anaerobic medium (GAM) agar plates (Nissui Pharmaceutical Co., Ltd., Tokyo, Japan) containing 30 µg mL⁻¹ spectinomycin. The insertional gene disruption was verified by Sanger sequencing following PCR amplification of the targeted gene locus. The primer pairs used are indicated in Supplementary Table 8.

The *BL105A_0502*-complemented strain was generated by introducing *BL105A_0502* complementation plasmid into the *BL105A_0502* mutant. The complementation plasmid was constructed by inserting the PCR-amplified DNA fragments corresponding to the *BL105A_0500* promoter region (605006 to 605305 nt) in GenBank accession no. NZ_AP014658.1[40] and promoter-less *BL105A_0502* gene (608449 to 609929 nt) into NdeI site of an *E. coli*–*Bifidobacterium* shuttle vector pTK2868 in this order. pTK2868 is a derivative of pTK2064[41] with the chloramphenicol-resistance gene replaced with the codon-optimised, *dnaA* promoter-driven version, which was synthesised by GenScript Inc. (Piscataway, NJ, USA). The replacement was carried out using HindIII sites of pTK2064. The primer pairs and the synthesised sequence used are shown in Supplementary Tables 8 and 9, respectively.

**Homologue search and phylogenetic analysis**. A BLASTP search was performed with the Bal6GBP sequence[25] as the query against all ORFs of the *B. longum* subsp. *longum* 105-A strain. To analyse the gene landscape in the genomes of lactulose-assimilating 10 bifidobacterial strains, *BL105A_0495-0502* sequences were used as queries (Fig. 3, Supplementary Fig. 1), using an e-value threshold <10⁻¹⁰⁰. Prevalence of *BL105A_0502* homologue was also analysed using a BLASTP search against the refseq protein database and all publicly available genomes of *Bifidobacterium* (1033 genomes, https://ftp.ncbi.nlm.nih.gov/genomes/refseq/bacteria/), with identity and query coverage threshold of >60 (%). The strains without species identification were omitted from the analysis.

The protein sequences of ten *BL105A_0502* homologues listed in Fig. 4 were aligned by the MUSCLE alignment tool (MUSCLE v3.8.31) with default settings. After being trimmed using Gblocks version 0.91b, the phylogenetic tree was generated using maximum likelihood in PhyML 3.3. The consensus tree was computed using the Consense module from Phylip package v3.69 using the majority rule method.

**Quantification of the LT-SBP**. Specific primer pairs were designed for each subtype of LT-SBP based on the aligned sequences by multiple sequence alignment using fast Fourier transform (MAFFT)[42] (Supplementary Table 4). The specificity of the primer pairs was confirmed by both BLASTN alignments performed against all genomes of publicly available bacteria and PCR using DNA samples extracted from all strains applied in this study. Real-time PCR was performed in a total volume of 20 µl per reaction using TB Green® Premix Ex Taq™ Tli RNaseH Plus (Takara Bio Inc., Shiga, Japan) with the following protocol: preheating at 95 °C for 20 s, followed by 40 cycles of denaturation at 94 °C for 3 s, annealing and extension at 60 °C for 30 s. All samples were assessed in duplicate. The detection limit for all real-time PCR assays was 10⁶ copies g⁻¹ faeces. The cell number of *Bifidobacterium* was previously measured in the clinical study[9].

**Metagenomic analysis**. Danish metagenomic data were obtained from NCBI Sequence Read Archive (PRJEB2054)[28]. After removing raw reads containing degenerate base (N), remained reads containing quality values of 17 or less were consecutively tailed-cut at the 3′ termini within the cutadapt programme (version 2.9)[43]. Read length with 50 or less base pairs were subsequently discarded. Sequences consistent with data from the Genome Reference Consortium human build 38 (GRCh38) and phiX reads were removed by mapping with Bowtie2 (version 2.3.4.1)[44].

Japanese metagenomic data were obtained from DDBJ Sequence Read Archive (DRP003048)[27]. The obtained sequences were quality-trimmed and quality-filtered with custom Perl and Python scripts and publicly available software as follows. After removing sequences less than 60 nucleotides or containing degenerate bases (N), all duplicates (a known artefact of pyrosequencing), defined as sequences in which the initial 20 nucleotides are identical and that share an overall identity of more than 97% throughout the length of the shortest read, were removed by CD-HIT-454. Host DNA was removed by mapping to the GRCh38 with Burrows-Wheeler Aligner (BWA) algorithm using the maximal exact match (MEM) option.

The occurrence of the *BL105A_0502* homologues gene (identity > 65%, read coverage > 60%) in the Danish and Japanese cohorts was examined by using the TBLASTN algorithm (BLAST + v2.6.0) against the quality-filtered reads described above. A ratio of the LT-SBP reads to total reads greater than 10⁻⁶ was judged as LT-SBP positive subjects.

**Statistics and reproducibility**. Statistical analyses were performed using R software version 3.6.0[45]. Differences in the cell number of *Bifidobacterium* and the total copy number of LT-SBPs before and after ingestion were analysed by a two-tailed paired student's *t*-test. A Pearson correlation test was used to determine any correlation. Samples were grouped based on the total copy number of LT-SBPs prior to lactulose ingestion, and intergroup differences of the fold change in the cell

number of *Bifidobacterium* were analysed with Tukey's test using the R package "multcomp"[46].

**Reporting summary**. Further information on research design is available in the Nature Research Reporting Summary linked to this article.

## Data availability

Source data for figures are provided with this paper as Supplementary Data. Any remaining information can be obtained from the corresponding author upon reasonable request.

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

## Acknowledgements

We thank Aya Mizuno and Toshihiko Katoh for providing technical support for these experiments, Miriam Nozomi Ojima for carefully proofreading the manuscript, Satoru Fukiya and Atsushi Yokota for providing the *Bifidobacterium* gene manipulation tool, and Yasunobu Kano and Tohru Suzuki for providing B. longum subsp. longum 105-A. This study was supported, in part, by Grants-in-Aid from the Institute for Fermentation, Osaka (K-25-04 to T.K., S.K., and M.S.).

## Author contributions

K.Y., Y.S., H.I., J.X., T.K., and T.O. contributed to the conception and design of the study; R.H., M.S., S.K., and T.K. designed and constructed the SBP mutants. M.C., M.S., and T.K. constructed the complementation plasmid. Y.S. and HI conducted the clinical study for lactulose ingestion. KY performed the statistical analysis; K.Y., T.K., and T.O. interpreted the data for the study; K.Y. and T.O. wrote the first draft of the manuscript; and K.Y., M.S., and T.O. wrote sections of the manuscript. All authors contributed to manuscript revision and read and approved the submitted version.

## Competing interests

Authors K.Y., Y.S., J.X., and T.O. are employees of Morinaga Milk Industry Co., Ltd. Employment of M.S. at Kyoto University is supported by Morinaga Milk Industry Co., Ltd. The remaining authors declare no competing interests.
