## [Peer Review File · Communications Biology]

Reviewers' Comments:

Reviewer #1:

Remarks to the Author:

In manuscript "Gut microbiota response to lactulose ingestion relies on a solute-binding protein-dependent ABC transporter from *Bifidobacterium*", Yoshida and coworkers described a locus (BL105A_0502) potentially involved in lactulose uptake. The number of copies of this locus per gram of faeces before lactulose ingestion shows a marked increase after lactulose ingestion in a subset of individuals. Based on these results, the authors suggest that prebiotics responders and non-responders can be identified.

This reviewer appreciates and values the amount of work done and is aware of the effort made by the authors to carry out this study. However, I consider that the work has important weaknesses that must be addressed before considering the manuscript for publication.

Some of my main concerns, and other minor comments, are included below:

Main weaknesses.

1. By inactivating the locus BL105A_0502, the mutant showed impaired growth compared with the other strains. This is a significant result showing the potential involvement of the locus in lactulose growth. However, the mutant should be complemented (with the BL105A_0502 locus included in a plasmid for instance) to regain the WT phenotype. I strongly recommend to perform this experiment.

2. In fig 4 you use a selection of strains from different species. Is the presence of BL105A_0502 a common feature in other strains/other species? There are many available *Bifidobacterium* genomes and I recommend to carry out a bioinformatic analysis to show the ubiquity of the locus in these and other species and several strains of the same species.

3. When testing the effect of lactulose ingestion on LT-SBP abundance, you just mention *Bifidobacterium*. But do you have additional information about the different *Bifidobacterium* species increasing their numbers in lactulose? This information, if available, should be included in the MS.

Minor comments:

In the abstract: "These results enable us to understand the molecular basis ... and provide a mechanistic insight into..." The results do not clarify any mechanisms, this phrase should be changed.

In the introduction: "we have identified the transport system responsible for lactulose assimilation by *Bifidobacterium* species". You overestimate your results (this is just one example). You have not characterised lactulose assimilation. You have not studied different *Bifidobacterium* species. Consider rephrasing the sentence and review the text to soften your conclusions. In other parts of the text you mention that the locus is responsible for the uptake of lactulose (you have not demonstrated it).

B. infantis JCM1222 has not transmembrane domains (Suppl. Fig. 1) but it grows very well in lactulose (Fig. 4). How can you explain this?

L107-115. This section is not properly connected with the rest of the manuscript.

L144-145. *B. longum* subsp. *longum* has a wide age distribution not only in Japanese subjects...

L149: This was possibly because there was limited physical space for a further increase in the *Bifidobacterium* abundance in the competitive ecosystem of adult intestines... You have not proved this.

L155-156: We speculate that certain growth inhibitors, such as bacteriocin, Bacteriophages and antimicrobial peptides produced by other bacteria and/or hosts, may exist in the guts of these subjects, which consequently prevent the growth of *Bifidobacterium*. Too much speculation, you do not have any clue about it.

Reviewer #2:

Remarks to the Author:

1. Brief summary of the manuscript

This paper studies the response and increase of Bifidobacterium to lactulose ingestion, by focusing on a lactulose transporter.

2. Overall impression of the work

The manuscript is well written and easy to follow. However, after a careful reading of the manuscript, I do not think that the present manuscript would be acceptable for Nature Communications Biology, at least in the present form. I hope the following suggestion would be helpful to the authors to improve their work.

3. Specific comments, with recommendations for addressing each comment

There are only few aspects that can be improved by adding some additional information to make it even more easy to read. They are commented below:

-Abstract section:

I would change the first sentence in order to avoid this "our previous study...". Instead, authors can delete the first sentence and they can use: "This study investigates the response of Bif to lactulose ingestion in healthy Japanese subjects, focusing on a lactulose transporter".

-Introduction: In Line 22, you mentioned "lactulose galacto-oligosaccharides" to be GOS. In my opinion, GOS are referred only to lactose galacto-oligosaccharides, so a short note should be included to clarify that these specific galacto-oligosaccharides are those containing lactulose instead of lactose. Line 43 also refers to "GOS"... are they derived from lactose or lactulose? Line 26: add references for Works studying lactulose as promoting increase of Bif in human gut.

-Results section:

Line 51, "Results" should be in bold, probably.

This work studies the response and increase of Bifidobacterium to lactulose ingestion, authors should not include results from "their previous clinical study" as results here. They can discuss results, but not include them and use figures regarding other publication. I suggest analyse this aspect and delete the corresponding associated figures (as Fig 1a, for example). The same applies for Fig2, although I could be wrong, so please analyse this and explain for better comprehension. Line 77, avoid "our" and use "In a previous study...". Furthermore, is "Sakai, under review" the same as cited as number 9 or 17 in the list of references? I guess they refer to another work, but as no information is given I doubt they can include as it is (maybe this could be solved by including the journal they submitted the manuscript).

Methods

Line 172, "Methods" should be in bold.

For faecal DNA samples analysed... Healthy Japanese subjects were donors... Could authors give some more information about number of donors/samples, age range, only performed in women and no men (or this situation was in "their" previous study?), and so on?

Bacteria and culture condition: Explain why only "long times" was considered, and not 5 or 8h was selected. It seems that with more time points better understanding would have been obtained.

Why did they decide that an OD>0.3 is a positive growth in the medium? Is it probably because ongoing studies in the laboratory have demonstrated that?

Figures: Maybe they can combine information contained in Figure 4 with that presented in Table 1. I cannot distinguish whether information in Figure 6 combines results obtained in this work with other data from previous studies done by the group... Furthermore, information in lines 390-394 can be added to "Results" section because I feel there is too information in this footnote.

I think that short chain fatty acid analysis could be positive to strengthen the conclusions of this study. However, this is a decision of authors, regarding availability of equipment and so on.

Finally, in my opinion, a claim of the paper is that the authors found that the number of LT-SBPs could be a marker for predicting subjects with the potential for a marked increase in bifidobacterial abundance upon lactulose intake, which can mark potential nutritional interventions in the future.

Reviewer #3:

Remarks to the Author:

Yoshida et al., present a manuscript investigating the role of an ABC transporter in lactulose metabolism and its putative role in enriching the gut microflora with bifidobacteria that encode this transporter. This is generally timely and of interest, and builds on relevant recently published studies.

Some of the narrative framing should be re-assessed in light of the previously published literature and to more accurately reflect the results presented in the manuscript. This starts with the title, and carries throughout the text, for which the authors should rather consider a narrative centered on the role of an ABC transporter in bifidobacteria responsible for enrichment following the ingestion of lactulose. The paper reads more like the characterization of an ABC transporter involved in intestinal bifidobacterial enrichment following lactulose consumption (there is no gut microflora response *per se*, nor characterization of the SBP). Also, this story build on previously published studies showing the impact of lactulose consumption, including the identification of key elements involved in bifidobacteria, so impact and novelty should be presented accordingly. Given the role of some of the authors in the published literature, this is important.

The most critical aspect of the paper is perhaps the identification and characterization of the lactulose ABC transporter, presumably involved in lactulose uptake by bifidobacteria. The authors must explain why and how it was identified more clearly and then characterize the various elements of the whole ABC transporter accordingly. Without actually presenting any data about uptake and transport, the authors need to rely on a more comprehensive and convincing set of *in silico* (and perhaps more) analyses showing what the various elements of the transporter are, and how they compare and contrast to others. Absent actual transport studies, the verbiage needs to be toned down and altered to “predictably, presumably, potentially, likely” relay a transport and uptake role. The phenotype of the engineered variant is noted and convincing. The most intriguing data presented is in Bif increase post ingestion and the distinction between responders and non-responders, so the authors should consider building the narrative on this early on and then characterize the predicted ABC transporter. The authors could also consider complementing the genomics data with transcriptomics or proteomics or metabolomics to more comprehensively and convincingly support their argument. Given the previous publication of several studies (refs 18-20) that implicate the uptake of GOS by bifidobacteria, proper framing in light of and complementation of the previously established literature is important, so the readership can properly assess impact and novelty.

Some of the conclusions presented by the authors need to be toned down or reframed to more accurately be supported by the presented data. For instance, no “assimilation” not “transport” is actually shown and it is unclear whether “the results of this study show lactulose is a highly effective prebiotic”, absent any efficiency data *per se*.

As mentioned by the authors, similar studies have identified how FOS and GOS prebiotics are uptaken and catabolized by probiotic bacteria. Besides the impact on the microbiome, the authors should also look at and cite and discuss these kinds of studies to build a more convincing and comprehensive argument on the ABC transporters presumably involved in lactulose uptake. In particular, the authors should provide comparative genomics data on how widespread (within and beyond bifidobacteria) this transporter is and how it compares and contrasts to previously identified carbohydrate transporters previously identified in bifidobacteria and other probiotics.

While investigating the occurrence of sequences of interest within the Japanese cohorts is relevant and justified, it would be desirable to complement this study to show how widespread (or not) these transporters are in other populations / cohorts / datasets, and whether patterns of interest arise (e.g. linked to age, diet or other factors of interest).

Figure 2 needs to be expanded to more genomes and species of interest and elements therein need to be re-arranged to be more visually clear and compelling. At the very minimum, genes should be flipped to be aligned (stacked) and additional panels need to be provided to provide more granularity and details about ABC transporter elements and key motifs and annotations and conservations, and perhaps the authors could consider a heat map to show levels of amino acid conservations for all genes of interest across bifidobacterial species of common interest (at least for organisms discussed in figure 4 and figure 5 and table 1). The authors could also complement this with transcriptional data to show induction in the presence of lactulose and show loss in the knock out, and perhaps more.

It is unclear why the timeline is so limited in figure 3, with cells seemingly still in the log phase before the experiment is terminated.

Results for Figure 4 should be re-ordered either in logical results-based patterns (increasing or decreasing OD patterns) or per the organisms phylogeny (16S-based order), or per relevant data (see suggested heat map to expand figure 2).

COMMSBIO-20-1630A

Reviewer's comment

Reviewer #1

In manuscript "Gut microbiota response to lactulose ingestion relies on a solute-binding
protein-dependent ABC transporter from Bifidobacterium", Yoshida and coworkers described a locus
(*BL105A_0502*) potentially involved in lactulose uptake. The number of copies of this locus per
gram of faeces before lactulose ingestion shows a marked increase after lactulose ingestion in a
subset of individuals. Based on these results, the authors suggest that prebiotics responders and
non-responders can be identified.

This reviewer appreciates and values the amount of work done and is aware of the effort made by the
authors to carry out this study. However, I consider that the work has important weaknesses that
must be addressed before considering the manuscript for publication.

Some of my main concerns, and other minor comments, are included below:

Main weaknesses.

By inactivating the locus *BL105A_0502*, the mutant showed impaired growth compared with the
other strains. This is a significant result showing the potential involvement of the locus in lactulose
growth. However, the mutant should be complemented (with the *BL105A_0502* locus included in a
plasmid for instance) to regain the WT phenotype. I strongly recommend to perform this experiment.

> Thank you very much for your suggestion. We have constructed a plasmid-based complementation
strain and evaluated its growth on lactulose. As expected, the complementation strain with
*BL105A_00502* showed a growth profile comparable to the WT strain. This result and corresponding
sentence were added in Fig.2c, d and L67-71.

Fig. 2. Growth profiles of *B. longum* subsp. *longum* 105-A wild-type and insertional mutant strains

in modified MRS medium supplemented with lactose (a) or lactulose (b) as the sole carbon source.
Growth profiles of *B. longum* subsp. *longum* 105-A wild-type, *BL105A_0502* insertional mutant
strain, and *BL105A_0502* complemented strain in modified MRS medium supplemented with lactose
(c) or lactulose (d) as the sole carbon source. OD_{600nm} values were determined at the indicated time
points. The presented data are the mean ± SD of at least three independent assays.

In fig 4 you use a selection of strains from different species. Is the presence of BL105A_0502 a
common feature in other strains/other species? There are many available Bifidobacterium genomes
and I recommend to carry out a bioinformatic analysis to show the ubiquity of the locus in these and
other species and several strains of the same species.

> We appreciate your kind suggestions. Following your advice, we have conducted a BLASTP
search against all publicly-available genomes of *Bifidobacterium* (1,033 genomes in total) to
investigate how the LT-SBP homologue was conserved in the strains/species of *Bifidobacterium*.
Fifty-seven of 77 bifidobacteria species possessed the LT-SBP homologue as a core gene. Another
BLASTP search against the refseq protein database showed that a few of other genera were also
predicted to possess the LT-SBP homologue gene. The results are described in L89–94 and
Supplementary Tables S2 and S3.

We did not perform clustering analysis (like Fig.4) based on the LT-SBP homologues obtained from
1,033 bifidobacteria genome and refseq, because we do not have any data showing growth in
lactulose-supplemented medium. Additionally, the species shown in Fig. 4 are dominant
bifidobacteria in the adult human gut.

When testing the effect of lactulose ingestion on LT-SBP abundance, you just mention
Bifidobacterium. But do you have additional information about the different Bifidobacterium species
increasing their numbers in lactulose? This information, if available, should be included in the MS.

> In a previous clinical study, lactulose ingestion was shown to increase the abundance of *B.*
*catenulatum* group, *B. longum* group, *B. adolescentis* and *B. breve* in the faeces. We would like to
send a PDF file of a paper that describes the above data, which was recently published
(doi:10.3920/BM2020.0100). Information as a reference has been revised (L74–76)

Minor comments:

In the abstract: “These results enable us to understand the molecular basis ... and provide a
mechanistic insight into... “The results do not clarify any mechanisms, this phrase should be
changed.

>Thank you for your suggestion. We have revised the sentence as follows:

“These results help predict the prebiotics-responder and non-responder status and provide an insight

into clinical interventions that test the efficacy of prebiotics.” (L8–10)

In the introduction: “we have identified the transport system responsible for lactulose assimilation by
*Bifidobacterium* species”. You overestimate your results (this is just one example). You have not
characterised lactulose assimilation. You have not studied different *Bifidobacterium* species.
Consider rephrasing the sentence and review the text to soften your conclusions. In other parts of the
text you mention that the locus is responsible for the uptake of lactulose (you have not demonstrated
it).

> Thank you for your suggestion. We have rephrased our conclusion as follows:

“In this study, we have identified the SBP possibly responsible for lactulose uptake by
*Bifidobacterium longum* subsp. *longum* 105-A, as a representative of lactulose-utilising strain.”
(L47–48).

*B. infantis* JCM1222 has not transmembrane domains (Suppl. Fig. 1) but it grows very well in
lactulose (Fig. 4). How can you explain this?

> Thank you for your insightful comment.

Not all orphan SBP genes encoded on bacterial genomes are remnants of nonfunctional ABC
transporters (Chen et al. 2009, reference No. 30). We speculate that the orphan LT-SBP works as a
functional ABC transporter with transmembrane domain encoded on different regions of the genome.
A sentence was added in the discussion (L151–155).

L107-115. This section is not properly connected with the rest of the manuscript.

> The following sentence was added in the first part of this section.

To predict the rate of potential “high responders” in Japanese cohort, we assessed ... (L114).

L144-145. *B. longum* subsp. *longum* has a wide age distribution not only in Japanese subjects...

> We have removed the word “Japanese” and added two references reporting in those results in
different populations (L159–161).

L149: This was possibly because there was limited physical space for a further increase in the
*Bifidobacterium* abundance in the competitive ecosystem of adult intestines... You have not proved
this.

> We have removed the sentence and added other speculations as follows:

“This was possibly because of dilution effect, in which limited lactulose availability could mask the
response of bifidobacteria to the prebiotic.” (L168–169)

L155-156: We speculate that certain growth inhibitors, such as bacteriocin, Bacteriophages and

antimicrobial peptides produced by other bacteria and/or hosts, may exist in the guts of these
subjects, which consequently prevent the growth of Bifidobacterium. Too much speculation, you do
not have any clue about it.

>We have removed this sentence.

Again, thank you for allowing us to strengthen our manuscript with your valuable comments and
queries. We have worked hard to incorporate your feedback and hope that these revisions persuade
you to accept our submission.

Sincerely,

Reviewer #2

Specific comments, with recommendations for addressing each comment

There are only few aspects that can be improved by adding some additional information to make it
even more easy to read. They are commented below:

-Abstract section:

I would change the first sentence in order to avoid this “our previous study...”. Instead, authors can
delete the first sentence and they can use: “This study investigates the response of Bif to lactulose
ingestion in healthy Japanese subjects, focusing on a lactulose transporter”.

>Thank you very much for your kind suggestion. Based on your sentence, we have revised the
sentence as follows:

“This study aims to understand the mechanistic basis underlying the response of Bifidobacterium to
lactulose ingestion in guts of healthy Japanese subjects, with specific focus on a lactulose
transporter.” (L1-2)

-Introduction:

In Line 22, you mentioned “lactulose galacto-oligosaccharides” to be GOS. In my opinion, GOS are
referred only to lactose galacto-oligosaccharides, so a short note should be included to clarify that
these specific galacto-oligosaccharides are those containing lactulose instead of lactose. Line 43 also
refers to “GOS”... are they derived from lactose or lactulose?

>We apologize for the careless typo. We added “,” after the word “lactulose” as follows:

“In addition to oral treatment with probiotic bifidobacteria, some oligosaccharides, such as lactulose,
galacto-oligosaccharide (GOS) and fructo-oligosaccharide (FOS) ...”(L19)

Line 26: add references for Works studying lactulose as promoting increase of Bif in human gut.

> References were added (L22).

-Results section:

Line 51, “Results” should be in bold, probably.

>Revised (L53).

This work studies the response and increase of Bifidobacterium to lactulose ingestion, authors
should not include results from “their previous clinical study” as results here. They can discuss
results, but not include them and use figures regarding other publication. I suggest analyse this
aspect and delete the corresponding associated figures (as Fig 1a, for example). The same applies for
Fig2, although I could be wrong, so please analyse this and explain for better comprehension.

>Thank you very much for your suggestion. The results from the previous study were removed from
Fig 1. All the other data reported were obtained in this study without fold increase in cell number of

LT-SBP in Fig 5b, c. We revised sentences throughout the manuscript.

Line 77, avoid “our” and use “In a previous study...”. Furthermore, is “Sakai, under review” the same
as cited as number 9 or 17 in the list of references? I guess they refer to another work, but as no
information is given I doubt they can include as it is (maybe this could be solved by including the
journal they submitted the manuscript).

>We apologize for the confusion caused. We referred to three independent papers whose reference
numbers are 9, 19 and 26 reported by the same first author. The reference 26 has been recently
published (doi:10.3920/BM2020.0100), so we would like to send the PDF file. Corresponding
descriptions and references were also revised (L34–40, 74–76, 353–355).

Line 172, “Methods” should be in bold.

>Revised (L198).

For faecal DNA samples analysed... Healthy Japanese subjects were donors... Could authors give
some more information about number of donors/samples, age range, only performed in women and
no men (or this situation was in “their” previous study?), and so on?

>Thank you for your helpful comment. We have added the subject information in the method section
(L200–204).

Bacteria and culture condition: Explain why only “long times” was considered, and not 5 or 8h was
selected. It seems that with more time points better understanding would have been obtained.

> Thank you for your helpful suggestion.

We monitored the OD at 8 different time points (0, 4, 16, 20, 24, 32, 40 and 48) in the revised
manuscript (please see Fig. 2).

Why did they decide that an $OD > 0.3$ is a positive growth in the medium? Is it probably because
ongoing studies in the laboratory have demonstrated that?

> We regarded the OD of the *BLI05A_0502* mutant at 24 hours as limited growth. As shown in Fig.
2b, the OD was between 0.2 and 0.3.

Fig. 2. Growth profiles of *B. longum* subsp. *longum* 105-A wild-type and insertional mutant strains in modified MRS medium supplemented with lactose (a) or lactulose (b) as the sole carbon source.

Growth profiles of *B. longum* subsp. *longum* 105-A wild-type, *BL105A_0502* insertional mutant strain, and *BL105A_0502* complemented strain in modified MRS medium supplemented with lactose

(c) or lactulose (d) as the sole carbon source. OD_{600nm} values were determined at the indicated time points. The presented data are the mean ± SD of at least three independent assays.

Figures:

Maybe they can combine information contained in Figure 4 with that presented in Table 1.

>Thank you for your advice. We removed Table 1 and the data (e.g., accession No.) was combined with supplementary Fig.2.

I cannot distinguish whether information in Figure 6 combines results obtained in this work with other data from previous studies done by the group...

>Only the data of *Bifidobacterium* abundance was used from the previous study. To avoid confusion, information was added in the method section (L267–268).

Furthermore, information in lines 390–394 can be added to “Results” section because I feel there is too information in this footnote.

>We have removed the information from the footnote and revised the corresponding sentence (L107–108)

I think that short chain fatty acid analysis could be positive to strengthen the conclusions of this study. However, this is a decision of authors, regarding availability of equipment and so on.

>Thank you very much for your suggestion. We measured the concentration of acetic acid during the culturing experiment (Supplementary Fig.1). As you mentioned, we think the data strengthens our conclusion.

Supplementary Fig 1. Acetic acid concentration at the indicated time points in modified MRS
 medium supplemented with lactulose as the sole carbon source. The presented data are the mean \pm
 SD of at least three independent assays.

Finally, in my opinion, a claim of the paper is that the authors found that the number of LT-SBPs
 could be a marker for predicting subjects with the potential for a marked increase in bifidobacterial
 abundance upon lactulose intake, which can mark potential nutritional interventions in the future.

>Thank you for your positive comment. We fully agree with you and hope to lead nutritional
 interventions in the near future.

Again, thank you for allowing us to strengthen our manuscript with your valuable comments and
 queries. We have worked hard to incorporate your feedback and hope that these revisions persuade
 you to accept our submission.

Sincerely,

Reviewer #3

Some of the narrative framing should be re-assessed in light of the previously published literature
and to more accurately reflect the results presented in the manuscript. This starts with the title, and
carries throughout the text, for which the authors should rather consider a narrative centered on the
role of an ABC transporter in bifidobacteria responsible for enrichment following the ingestion of
lactulose. The paper reads more like the characterization of an ABC transporter involved in intestinal
bifidobacterial enrichment following lactulose consumption (there is no gut microflora response per
se, nor characterization of the SBP).

>We appreciate your kind review and many helpful suggestions. We have carefully amended
sentences throughout the manuscript. In the beginning, we have changed the title to
“*Bifidobacterium* response to lactulose ingestion relies on a solute-binding protein-dependent ABC
transporter”.

Also, this story build on previously published studies showing the impact of lactulose consumption,
including the identification of key elements involved in bifidobacteria, so impact and novelty should
be presented accordingly. Given the role of some of the authors in the published literature, this is
important.

> Thank you very much for your comment. We have explained the findings of the previous study in
more detail (L34–41).

The most critical aspect of the paper is perhaps the identification and characterization of the
lactulose ABC transporter, presumably involved in lactulose uptake by bifidobacteria.

The authors must explain why and how it was identified more clearly and then characterize the
various elements of the whole ABC transporter accordingly. Without actually presenting any data
about uptake and transport, the authors need to rely on a more comprehensive and convincing set of
in silico (and perhaps more) analyses showing what the various elements of the transporter are, and
how they compare and contrast to others. Absent actual transport studies, the verbiage needs to be
toned down and altered to “predictably, presumably, potentially, likely” relay a transport and uptake
role. The phenotype of the engineered variant is noted and convincing.

> You have raised an important point; however, we had previously tried to identify the key gene
responsible for the utilisation of lactulose by bioinformatic methods only, and we did not obtain any
critical evidence from the analysis. For this reason, we have worked to identify the key gene by two
steps. We first performed BLASTP analysis using a query sequence of *Bal6GBP*, which has been
shown to bind to lactulose, followed by the construction of SBP mutants, which revealed that

*BL105A_0502* plays an important role in lactulose incorporation.

Based on your comments, we have constructed a complementary mutant with the *BL105A_0502*
gene. The complementary strain grew well in the medium with lactulose, indicating its ability to
transport lactulose.

We understand our study lacks biochemical data demonstrating the role of LT-SBP in lactulose
uptake. Initially (before submitting the original version), we conducted surface resonance plasmon
analysis to detect the binding of LT-SBP to lactulose, but no apparent signal was obtained under the
tested conditions. The results, however, do not indicate that the SBP is not involved in the lactulose
uptake. Recently, we observed a similar discrepancy between in vitro and in vivo data (Sakanaka et
al., reference No. 37), In that paper, FL1-BP (Fucosyllactose-binding protein 1) did not show
binding to 3'-fucosyllactose (3'-FL) while it binds to 2'-fucosyllactose (2'-FL). Nonetheless, FL1
transporter-expressing recombinant strain imports both 3'-FL and 2'FL at a similar rate in vivo (the
host strain was of course negative for both substrates). Therefore, it is not surprising that LT-SBP can
serve as lactulose transporter in vivo. We think growth profiles observed for WT, the insertion
mutant, and the complemented strain can convince readers that LT-SBP plays an important role in
lactulose assimilation.

We carefully checked throughout the draft manuscript, and some descriptions were rephrased (L47,
149, 193).

The most intriguing data presented is in Bif increase post ingestion and the distinction between
responders and non-responders, so the authors should consider building the narrative on this early on
and then characterize the predicted ABC transporter.

>Thank you for the helpful comment.

We have built the narrative on this earlier in the manuscript (L34–41).

The authors could also consider complementing the genomics data with transcriptomics or
proteomics or metabolomics to more comprehensively and convincingly support their argument.
Given the previous publication of several studies (refs 18-20) that implicate the uptake of GOS by
bifidobacteria, proper framing in light of and complementation of the previously established
literature is important, so the readership can properly assess impact and novelty.

>Thank you for your suggestion. We tried RNA-seq analysis but did not get expected results. It is
known that there are sometimes differences between gene and protein expressions in addition to
phenotype, however, we think the result by the complementary mutant (as mentioned above)
strongly supports our hypothesis.

Some of the conclusions presented by the authors need to be toned down or reframed to more

accurately be supported by the presented data. For instance, no “assimilation” not “transport” is
actually shown and it is unclear whether “the results of this study show lactulose is a highly effective
prebiotic”, absent any efficiency data per se.

>Our conclusion has been toned down (L47, 149–150, 193). Regarding the lactulose assimilation
and transport by *Bifidobacterium*, we could not directly characterize those mechanisms. Instead, we
measured the concentration of acetic acid and monitored the amount of lactulose remaining in the
culture (mMRS supplemented with lactulose). The *BL105A_0502* mutant showed a decrease in
acetic acid production and a delay in lactulose consumption compared to the wild-type strain (Fig. 2).
We argue that these data support the lactulose assimilation and transport in *Bifidobacterium longum*.

Fig. 2. Growth profiles of *B. longum* subsp. *longum* 105-A wild-type and insertional mutant strains
in modified MRS medium supplemented with lactose (a) or lactulose (b) as the sole carbon source.
Growth profiles of *B. longum* subsp. *longum* 105-A wild-type, *BL105A_0502* insertional mutant
strain, and *BL105A_0502* complemented strain in modified MRS medium supplemented with lactose
(c) or lactulose (d) as the sole carbon source. OD_{600nm} values were determined at the indicated time
points. The presented data are the mean ± SD of at least three independent assays.

As mentioned by the authors, similar studies have identified how FOS and GOS prebiotics are
uptaken and catabolized by probiotic bacteria. Besides the impact on the microbiome, the authors
should also look at and cite and discuss these kinds of studies to build a more convincing and
comprehensive argument on the ABC transporters presumably involved in lactulose uptake.

In particular, the authors should provide comparative genomics data on how widespread (within and
beyond bifidobacteria) this transporter is and how it compares and contrasts to previously identified
carbohydrate transporters previously identified in bifidobacteria and other probiotics.

> Thank you for your insight. We have referred to previous studies about GOS uptake and
catabolism and discussed the case of lactulose assimilation (L157–164).

We also have conducted a BLASTP search against all publicly-available genomes of
*Bifidobacterium* except for unidentified species (1,033 genomes in total) to investigate how the
LT-SBP homologue is conserved. Of 77 bifidobacteria species, 57 species possess the LT-SBP
homologue as a core gene for each species. The LT-SBP homologue was not coded in the genome of
the other 20 bifidobacterial species. Further BLASTP search against the refseq protein database
showed that some other genera, *Gardnerella vaginalis*, *Peptoniphilus lacrimalis* and *Lactobacillus*
*rhamnosus*, were also predicted to possess the LT-SBP homologous gene. The results are described
in L89–94 and Supplementary Table. 2 and 3.

While investigating the occurrence of sequences of interest within the Japanese cohorts is relevant
and justified, it would be desirable to complement this study to show how widespread (or not) these
transporters are in other populations / cohorts / datasets, and whether patterns of interest arise (e.g.
linked to age, diet or other factors of interest).

> Following your suggestion, we have investigated the distribution of the LT-SBP homologue in
other datasets: another Japanese cohort (Nishijima et al. 2016, reference No. 34) and a Danish cohort
(Qin et al. 2010, reference No. 35). The SBP homologues were found in over 80% of the Japanese
subjects but only 30% of the Danish subjects. The LT-SBP distribution was probably related with the
*Bifidobacterium* abundance in each subset. We speculate that this is because the relative abundance
of *Bifidobacterium* was higher in Japanese than in Danish (L185–192).

Figure 2 needs to be expanded to more genomes and species of interest and elements therein need to
be re-arranged to be more visually clear and compelling. At the very minimum, genes should be
flipped to be aligned (stacked) and additional panels need to be provided to provide more granularity
and details about ABC transporter elements and key motifs and annotations and conservations, and
perhaps the authors could consider a heat map to show levels of amino acid conservations for all
genes of interest across bifidobacterial species of common interest (at least for organisms discussed
in figure 4 and figure 5 and table 1). The authors could also complement this with transcriptional
data to show induction in the presence of lactulose and show loss in the knock out, and perhaps more.

>Thank you for your helpful suggestion. We have reflected your suggestion to flip the genes in Fig.1.
An expanded panel for 10 lactulose utilizing strains is shown in Supplementary Fig. 2. We did not
obtain transcriptional data for the complement because we have already proved its function using the
complementary strain with *BL105A_00502* gene (Fig. 2c, d).

 Supplementary Fig 2. Comparison of the genomic structure of lactulose-assimilating
 *Bifidobacterium* strains at the *BL105A_0502* locus. Solid arrows indicate open reading frames
 (ORFs) with their lengths proportional to the polypeptides chain lengths. The locus tags numbers are
 indicated inside the arrows. The amino acid identity (%) against homologue of *B. longum* subsp.
 *longum* 105-A is shown. The genes coding for transmembrane component, β -galactosidase, and SBP
 are coloured in black, grey, and light red, respectively.

 It is unclear why the timeline is so limited in figure 3, with cells seemingly still in the log phase
 before the experiment is terminated.

> Thank you for your helpful comment. We extended the timeline to 48h (Fig2.). This new figure
 clearly shows a significant delay in the growth of the *BL105A_0502* mutant.

 Results for Figure 4 should be re-ordered either in logical results-based patterns (increasing or
 decreasing OD patterns) or per the organisms phylogeny (16S-based order), or per relevant data (see
 suggested heat map to expand figure 2).

>We agree with you and have reordered Fig.3 by decreasing OD patterns

 Fig. 3. Growth capability of 14 strains belonging to 13 different *Bifidobacterium* species/subspecies
 in modified MRS medium supplemented with lactulose as the sole carbon source. Strains with an
 OD_{600nm} value greater than 0.3 after 24 h incubation was classified as a lactulose-assimilating
 phenotype (see Fig. 2b). The presented data are the mean \pm SD of at least two independent assays.

Again, thank you for allowing us to strengthen our manuscript with your valuable comments and
queries. We have worked hard to incorporate your feedback and hope that these revisions persuade
you to accept our submission.

Sincerely,

REVIEWERS' COMMENTS:

Reviewer #1 (Remarks to the Author):

The authors have properly addressed my previous concerns. My opinion is that the MS can be considered for publication in the current form.

Reviewer #2 (Remarks to the Author):

Authors improved their work based on reviewers' suggestions. Although I offered some suggestions to authors that would be helpful to them to improve their work, there are some questions I would like to be clarified.

- Materials and Methods: lactulose purity, being lactulose a synthetically-produced disaccharide (not mentioned throughout the manuscript), is not specified. Just include the information given in the previous study/studies ("lactulose crystal anhydrate powder (MLC-97, $\geq 97\%$, Morinaga Milk Industry Co., Ltd., Tokyo, Japan) was used", reference number 9). As monosaccharides or lactose impurities are below 3%, the demonstrated effect relies practically only on the disaccharide.

- Authors explained the findings of previous studies (L34–41) and stated that Lactulose ingestion increased the abundance of Bif in a 4.7 ± 8.6 -fold... Is this relevant taking into account that high standard deviation (and being $-8.6!!!$)? I checked Sakai et al. 2019 (10.3920/BM2018.0174), and I find information regarding the number of Bifidobacterium (log cfu/g faeces), being "higher during lactulose (9.53 ± 0.06) than placebo (9.16 ± 0.06) treatment" and that "the percentage of Bifidobacterium in faeces was also significantly higher during lactulose (25.3 ± 1.4) than placebo (18.2 ± 1.4) treatment". Moreover, concerning the percentage of Bifidobacterium, I cannot see why 25 is 4.7 four fold increase of 18. Maybe I am wrong, but I think this could be a misunderstanding for interested readers. Are they referring to Bif abundance or total copy number of LT-SBPs in samples?

-Line 104-107: I believe authors were trying to say "less than 107 or more than 109 copies per gram, $n = 4$ OR $n = 30$, respectively).

-Suggestion: revise the title of Fig 5. Change "obtained in the previous clinical trial of" by "after". Also in the text, other figures used parenthesis for the different graphs associated, for instance "(a)", "(b)", and so on. Please, check the use of these punctuation marks in order to write the text using the same criteria. They can add "refers to" here: "trial (#p refers to < 0.1 , *** $p < 0.001$, Tukey's test).".

Therefore, a minor revision is needed covering the current questions. The decision concerning acceptance of this contribution rely on the editor.

COMMSBIO-20-1630B

Reviewer's comment

Reviewer #1 (Remarks to the Author):

The authors have properly addressed my previous concerns. My opinion is that the MS can be
considered for publication in the current form.

**Thank you for taking the time to review our manuscript.**

Reviewer #2 (Remarks to the Author):

Authors improved their work based on reviewers' suggestions. Although I offered some suggestions
to authors that would be helpful to them to improve their work, there are some questions I would like
to be clarified.

- Materials and Methods: lactulose purity, being lactulose a synthetically-produced disaccharide (not
mentioned throughout the manuscript), is not specified. Just include the information given in the
previous study/studies ("lactulose crystal anhydrate powder (MLC-97, $\geq 97\%$, Morinaga Milk
Industry Co., Ltd., Tokyo, Japan) was used", reference number 9). As monosaccharides or lactose
impurities are below 3%, the demonstrated effect relies practically only on the disaccharide.

**We added the information to the Materials and Methods section as follows.**

**"(MLC-97), a commercially available product of Morinaga Milk Industry Co., Ltd (Tokyo, Japan)**
**with 97% purity, as determined by HPLC using a refractive index detector,...)" (L213–214)**

- Authors explained the findings of previous studies (L34–41) and stated that Lactulose ingestion
increased the abundance of Bif in a 4.7 ± 8.6 -fold... Is this relevant taking into account that high
standard deviation (and being -8.6 !!!)?

I checked Sakai et al. 2019 (10.3920/BM2018.0174), and I find information regarding the number of
Bifidobacterium (log cfu/g faeces), being "higher during lactulose (9.53 ± 0.06) than placebo
(9.16 ± 0.06) treatment" and that "the percentage of Bifidobacterium in faeces was also significantly
higher during lactulose (25.3 ± 1.4) than placebo (18.2 ± 1.4) treatment".

Moreover, concerning the percentage of Bifidobacterium, I cannot see why 25 is 4.7 four fold
increase of 18. Maybe I am wrong, but I think this could be a misunderstanding for interested
readers. Are they referring to Bif abundance or total copy number of LT-SBPs in samples?

**>Thank you for pointing out our errors. We should have calculated the mean value of the fold**
**change by geometric mean, but we had calculated it by arithmetic mean. In addition, the use of the**
**term "the abundance" at line 37 might have caused confusion. We intended to describe "the number**
**of Bifidobacterium cells before ingestion" there, not "percentage of Bifidobacterium." The sentence**

has been revised as follows:

“In a previous clinical study, which was a randomized, double-blind, placebo-controlled crossover
trial, we observed that two grams of lactulose ingestion per day for two weeks increased the number
of *Bifidobacterium* cells in the faeces of 49 healthy Japanese females with statistical significance
(2.2 (2.9)-fold increase, geometric mean (standard deviations))⁹.” (L36–38).

“... and the total copy number of the four SBP gene subtypes upon lactulose ingestion with
statistical significance after ingestion (2.2 (3.0)-fold, ...)” (L101)

The numbers in parentheses show geometric standard deviations.

-Line 104-107: I believe authors were trying to say “less than 107 or more than 109 copies per gram,
n = 4 OR n = 30, respectively).

>Thank you. We rewrote the sentence according to your suggestion (L108).

-Suggestion: revise the title of Fig 5. Change “obtained in the previous clinical trial of” by “after”.
Also in the text, other figures used parenthesis for the different graphs associated, for instance “(a)”,
“(b)”, and so on. Please, check the use of these punctuation marks in order to write the text using the
same criteria. They can add “refers to” here: “trial (#p refers to < 0.1, *** p < 0.001, Tukey’s test).”.

>Thank you for your helpful comments.

All of your suggestions have been incorporated in the revised manuscript (L458–465).

Therefore, a minor revision is needed covering the current questions. The decision concerning
acceptance of this contribution rely on the editor.

>Thank you for your valuable comments, which helped us significantly improve our manuscript. We
hope that the revision is satisfactory.

Sincerely,
